# Prostate MRI added to CAPRA, MSKCC and Partin cancer nomograms significantly enhances the prediction of adverse findings and biochemical recurrence after radical prostatectomy

Kevin Sandeman[1,2]*, Juho T. Eineluoto[2,3], Joona Pohjonen[2], Andrew Erickson[2], Tuomas P. Kilpeläinen[3], Petrus Järvinen[3], Henrikki Santti[3], Anssi Petas[3], Mika Matikainen[3], Suvi Marjasuo[4], Anu Kenttämies[4], Tuomas Mirtti[1,2], Antti Rannikko[2,3]

**1** Department of Pathology, University of Helsinki and Helsinki University Hospital, Helsinki, Finland,
**2** Research Program in Systems Oncology, Faculty of Medicine, University of Helsinki, Helsinki, Finland,
**3** Department of Urology, University of Helsinki and Helsinki University Hospital, Helsinki, Finland,
**4** Department of Diagnostic Radiology, Medical Imaging Center, University of Helsinki and Helsinki University Hospital, Helsinki, Finland

* kevin_sandeman@yahoo.se

## Abstract

### Background

To determine the added value of preoperative prostate multiparametric MRI (mpMRI) supplementary to clinical variables and their role in predicting post prostatectomy adverse findings and biochemically recurrent cancer (BCR).

### Methods

All consecutive patients treated at HUS Helsinki University Hospital with robot assisted radical prostatectomy (RALP) between 2014 and 2015 were included in the analysis. The mpMRI data, clinical variables, histopathological characteristics, and follow-up information were collected. Study end-points were adverse RALP findings: extraprostatic extension, seminal vesicle invasion, lymph node involvement, and BCR. The Memorial Sloan Kettering Cancer Center (MSKCC) nomogram, Cancer of the Prostate Risk Assessment (CAPRA) score and the Partin score were combined with any adverse findings at mpMRI. Predictive accuracy for adverse RALP findings by the regression models was estimated before and after the addition of MRI results. Logistic regression, area under curve (AUC), decision curve analyses, Kaplan-Meier survival curves and Cox proportional hazard models were used.

### Results

Preoperative mpMRI data from 387 patients were available for analysis. Clinical variables alone, MSKCC nomogram or Partin tables were outperformed by models with mpMRI for

**Data Availability Statement:** All relevant data are within the manuscript and its Supporting Information files.

**Funding:** JTE: Orion Research Foundation sr TM: Academy of Finland TM and AR: Cancer Society Finland and HUS Helsinki University Hospital JTE's salary was paid by the grant from the Cancer Society Finland. The funders had no role in study design, data collection and analysis, decision to publish, or preparation of the manuscript.

**Competing interests:** The authors have declared that no competing interests exist.

the prediction of any adverse finding at RP. AUC for clinical parameters versus clinical parameters and mpMRI variables were 0.77 versus 0.82 for any adverse finding. For MSKCC nomogram versus MSKCC nomogram and mpMRI variables the AUCs were 0.71 and 0.78 for any adverse finding. For Partin tables versus Partin tables and mpMRI variables the AUCs were 0.62 and 0.73 for any adverse finding. In survival analysis, mpMRI-projected adverse RP findings stratify CAPRA and MSKCC high-risk patients into groups with distinct probability for BCR.

## Conclusions

Preoperative mpMRI improves the predictive value of commonly used clinical variables for pathological stage at RP and time to BCR. mpMRI is available for risk stratification pre-biopsy, and should be considered as additional source of information to the standard predictive nomograms.

## Introduction

Prediction of prostate cancer (PC) extent and overall pretreatment risk assessment are main challenges in everyday urological practice. In addition to the pathological Gleason Grade Group (GGG) assessed in prostate biopsies (Bx), other variables such as age, prostate-specific antigen (PSA), clinical stage, and biopsy-based tumor volume have been used to predict post-operative tumor characteristics of an individual PC patient [1–5]. These variables have also been incorporated into mathematical models to construct tools such as Partin tables, Memorial Sloan Kettering Cancer Center (MSKCC) nomograms and Cancer of the Prostate Risk Assessment (CAPRA) score [2, 3, 5] in order to improve predictions. Recently, prostate multiparametric magnetic resonance imaging (mpMRI) has been suggested as a tool to improve PC diagnostics and especially of clinically significant PC [4]. Prostate Imaging Reporting and Data System (PI-RADS) was introduced in 2012 to standardize the reporting of mpMRI findings [6]. The use of the PI-RADS approach is based upon literature evidence and consensus expert opinions [6]. However, only a few studies have assessed the value of mpMRI to predict adverse pathology and biochemical recurrence (BCR) in men undergoing RP, and even fewer studies have evaluated the added value of MRI over the pre-existing clinical variables and nomograms, and the results have been conflicting [7–11]. Further, combination of mpMRI and predictive nomograms in outcome assessment have not been studied in detail.

The aim of this study was to evaluate the added value of mpMRI in comparison to commonly used clinical variables as well as standard pre- and postoperative nomograms to predict adverse pathology and BCR in men undergoing RP in a contemporary series utilizing PI-RADS reporting system [6, 12].

## Materials and methods

### Patient population

A total of 598 patients underwent robot assisted laparoscopic prostatectomy (RALP) at the HUS Helsinki University Hospital during the study period from January 2014 to September 2015. Of these, 211 (35%) patients underwent RALP without preoperative mpMRI and were excluded from this study. Thus, the study cohort consists of 387 consecutive patients that underwent mpMRI prior to their RALP at treating clinician's discretion. Preoperative clinical

and pathological data of the entire cohort were retrieved from the prospective RALP database. These data included age, clinical stage, PSA, primary GS, secondary GS, number of positive biopsies and total number of biopsies. These were markedly the same between the mpMRI cohort and non-mpMRI cohort, confirming that no obvious selection bias existed for patient referral to pre-operative mpMRI (Table 1). At the time of the data curation, we included all the patients having undergone MRI according to contemporary guidelines and having follow-up information in order to conduct survival analysis. Therefore, no formal power calculations were performed but, instead, all patients during the study period were included. The data was accessed and collected between October 2016 and June 2019 from the electronic health records of the HUS Helsinki University Hospital. The original data was accessed based on patients' social security numbers but in the study database all data was handled pseudonymized and according to the study approval by the HUS Helsinki University Hospital. The study was a retrospective registry study and thus no informed consent was required, based on the national and European Union legislation. The study was conducted according to the Declaration of Helsinki and approved by the Institutional Ethics Committee of Helsinki University Hospital (diary number 386/13/03/02/2014).

## mpMRI technique

Prostate mpMRI was performed without an endorectal coil using a 3.0 T Philips Achieva MRI scanner, which produced 3-mm thick image slices. The modality included T2WI, DWI with ADC mapping and DCE conforming to European Society of Urogenital Radiology guidelines as previously described [6, 13]. The mpMRI data were interpreted by one of the four experienced uroradiologists all of whom were familiar with prostate mpMRI i.e. each uroradiologist had read over 300 prostate mpMRIs per year. At the time, mpMRI data were reported according to PI-RADSv1 recommendations by using a structured formula as follows: number of lesions, tumor volume and maximal diameter, PI-RADS-score, extraprostatic extension (EPE), seminal vesicle invasion (SVI), node status (NS) and radiologic stage.

**Table 1. Preoperative variables of all RALP patients with and without preoperative mpMRI.**

|  | Preoperative mpMRI | | No preoperative mpMRI | |
|---|---|---|---|---|
|  | Result | IQR or % of total | Result | IQR of % of total |
| **Number of patients** | 387 | - | 211 | - |
| **Median age, yrs** | 65 | 60–69 | 66 | 61–70 |
| **Median preoperative PSA, ng/ml** | 9.0 | 6.2–13.6 | 8.3 | 3.6–9.9 |
| **Biopsy data available** | 384 | 99.2 | 211 | 100 |
| **GGG. No.** |  |  |  |  |
| **GGG1** | 90 | 23.3 | 46 | 21.8 |
| **GGG2** | 135 | 34.9 | 83 | 39.3 |
| **GGG3** | 105 | 27.1 | 52 | 24.6 |
| **GGG4** | 29 | 7.5 | 21 | 10.0 |
| **GGG5** | 25 | 6.5 | 9 | 4.3 |
| **Median prostate volume, cm$^3$** | 35 | 27–45 | - | - |
| **Median prostate weight, g** | 48 | 41–59 | - | - |

IQR: Interquartile Range, GGG: Gleason Grade Group

## Prostatectomy specimen and pathologic analysis

RALP was performed transperitoneally and the indication for extended lymphadenectomy was GS ≥4+3 or >5% risk of lymph node positive disease according to the Memorial Sloan Kettering nomogram [2, 14]. The prostate specimens and lymph nodes were processed as described earlier [15]. All prostate material was completely mounted. Lymphadenectomy specimens were mounted per localization. The original pathological diagnosis was made individually by 12 expert pathologists, out of which 4 reported 80% of the cases. Pathological anatomical diagnosis (PAD) report of RALP specimen including primary and secondary GS, tumor volume, margin status, EPE, SVI, NS and pathologic stage (pT) was collected from the RALP database [16]. Adverse pathologic findings were defined as EPE, SVI or positive NS.

## Statistical analyses

Multivariable logistic regression models were used to study the relationship between adverse RP findings (EPE, SVI or positive NS) and clinical variables available before RALP (preoperative PSA, age, cT-stage, percentage of positive biopsy cores, GGG), MSKCC preoperative nomogram and Partin Table estimates. Cox proportional hazards models and Kaplan-Meier survival curves were generated for clinical variables, CAPRA score and MSKCC nomogram parameters in predicting BCR. All models were compiled following the purposeful selection of covariates [17, 18]. Next, available mpMRI variables assessing EPE, SVI or positive NS, prostate volume and PI-RADS score were added to the models. The final models with and without the mpMRI variables were compared by decision curve analysis [19], area under the receiver operating characteristics (ROC) curve and multiparametric Wald test. Eighty-one patients had missing values for positive and total biopsy cores and these were imputed using multiple imputation by chained equations [20]. An alpha level of 0.05 was used for statistical significance. All statistical analyses were performed with R Statistical Software v.3.6.1 [21] using the packages: survival [22], precrec [23], mfp and mice[20].

## Results

### Cohort

Of the 387 mpMRI patients with a median age of 65 years at RALP, preoperative PSA and diagnostic biopsy data were available for 384 patients (Table 1). Distribution of mpMRI, clinical and pathological staging as well as PI-RADS scores and RP GGG are presented in S1 Table. The number of patients in each subcohort in the regression models and survival analyses are presented in S1 Fig. As Partin tables do not include cT3, patients with clinical T3 were excluded from the analysis concerning Partin tables. Similarly, as CAPRA is designed to predict BCR, it was omitted from the regression models for adverse RP findings.

### Predicting adverse findings at RP (Partin, MSKCC, clinical variables, mpMRI)

Results of the multivariable logistic regression models for any RP adverse findings are shown in Table 2. Values were dichotomized for clinical stage (<cT2 versus ≥cT3), GGG (<2 versus ≥3) and PIRADS score (<2 versus ≥3). From all considered clinical covariates, age was not significantly associated with any adverse finding. However, based on the known effect to PC risk, age was left in to the model. From mpMRI variables, prostate volume was not significantly associated with any adverse RP finding. All models with mpMRI variables were significantly different from models with clinical variables only based on the multiparametric Wald test (p-values < 0.001). From the preoperative clinical variables, cT-stage ≥ 3 had the highest odds

**Table 2. Regression model summaries: Prediction of any adverse findings at prostatectomy.**

| Model without mpMRI parameters | | | | Model with mpMRI parameters | | | |
|---|---|---|---|---|---|---|---|
| | Estimate | OR (95% CI) | *p* value | | Estimate | OR (95% CI) | *p* value |
| | | | | **MRI** | | | |
| - | - | - | - | MRI ANY | 1.420 | 4.14 (2.641–6.481) | **<0.001** |
| - | - | - | - | PI-RADS ≥ 3 | 1.218 | 3.40 (1.249–9.149) | **0.017** |
| - | - | - | - | MRI prostate volume | -0.013 | 0.99 (0.972–1.003) | 0.095 |
| **PARTIN** | | | | **PARTIN + MRI** | | | |
| Partin ANY | 0.026 | 1.03 (1.012–1.041) | **<0.001** | Partin ANY | 0.025 | 1.03 (1.011–1.039) | **<0.001** |
| - | - | - | - | MRI ANY | 1.277 | 3.59 (2.116–6.075) | **<0.001** |
| - | - | - | - | PI-RADS ≥ 3 | 0.837 | 2.31 (0.83–6.425) | 0.110 |
| - | - | - | - | MRI prostate volume | -0.010 | 0.99 (0.973–1.008) | 0.235 |
| **MSKCC** | | | | **MSKCC + MRI** | | | |
| MSKCC ANY | 0.053 | 1.05 (1.04–1.069) | **<0.001** | MSKCC ANY | 0.045 | 1.05 (1.03–1.063) | **<0.001** |
| - | - | - | - | MRI ANY | 1.139 | 3.12 (1.944–5.019) | **<0.001** |
| - | - | - | - | PI-RADS ≥ 3 | 1.090 | 2.97 (1.071–8.258) | **0.037** |
| - | - | - | - | MRI prostate volume | -0.017 | 0.98 (0.968–0.999) | **0.042** |
| **CLINICAL** | | | | **CLINICAL + MRI** | | | |
| PSAPre | 0.041 | 1.04 (1.012–1.073) | **0.008** | PSAPre | 0.052 | 1.05 (1.017–1.091) | **0.003** |
| Age | 0.036 | 1.04 (0.999–1.076) | 0.059 | Age | 0.144 | 1.16 (1.033–1.291) | **0.011** |
| GGG ≥ 3 | 0.508 | 1.67 (1.028–2.686) | **0.039** | GGG ≥ 3 | 0.521 | 1.68 (1.006–2.819) | **0.048** |
| Positive Bx(%) | 0.024 | 1.02 (1.012–1.036) | **<0.001** | % positive biopsies | 0.022 | 1.02 (1.01–1.034) | **<0.001** |
| cT ≥ 3 | 1.382 | 3.98 (2.029–7.817) | **<0.001** | cT ≥ 3 given MRI ANY | 2.047 | 7.74 (1.868–32.093) | **0.005** |
| - | - | - | - | cT ≥ 3 given MRI OC | 0.426 | 1.53 (0.659–3.56) | 0.323 |
| - | - | - | - | MRI ANY given cT ≥ 3 | 1.218 | 3.38 (1.939–5.893) | **<0.001** |
| - | - | - | - | MRI ANY given cT < 3 | -0.403 | 0.67 (0.14–3.187) | 0.614 |
| - | - | - | - | PI-RADS ≥ 3 | 1.104 | 3.02 (0.985–9.236) | 0.054 |
| - | - | - | - | MRI prostate volume | 0.184 | 1.20 (1.002–1.442) | **0.049** |
| - | - | - | - | Age x MRI prostate vol. | -0.003 | | **0.033** |
| - | - | - | - | cT ≥ 3 x MRI ANY | -1.621 | | 0.056 |

MRI: magnetic resonance imaging; ANY: suggestion of extraprostatic extension, seminal vesicle invasion or lymph node involvement; OC: organ-confined (≤pT2); cT: clinical stage

ratio of 3.98. In addition, The MSKCC nomogram and Partin Table risk predictions were significantly associated with any adverse RP finding (p < 0.001) with respective odds ratios of 1.05 and 1.03 (1% increase in risk score). In the ROC-AUC analyses, when mpMRI was added to the other models (clinical variables, Partin parameters and MSKCC estimates), all combined models outperformed the individual models without mpMRI. The decision curve analysis showed a net benefit for all models combined with mpMRI for patients with a threshold probability between approximately 10 to 15 and 55 to 75% for any adverse finding (Fig 1A–1F).

## BCR free survival (CAPRA, MSKCC, clinical variables, mpMRI)

Kaplan-Meier survival plots revealed that all preoperative clinical variables and mpMRI indicating adverse RP findings separate the patient cohort into two groups with different survival probability (for all variables p< 0.01 in log rank test, S2A–S2F Fig). Neither mpMRI prostate volume (> 35 cc) nor the PIRADS score higher than two separated the groups significantly (S3A and S3B Fig).

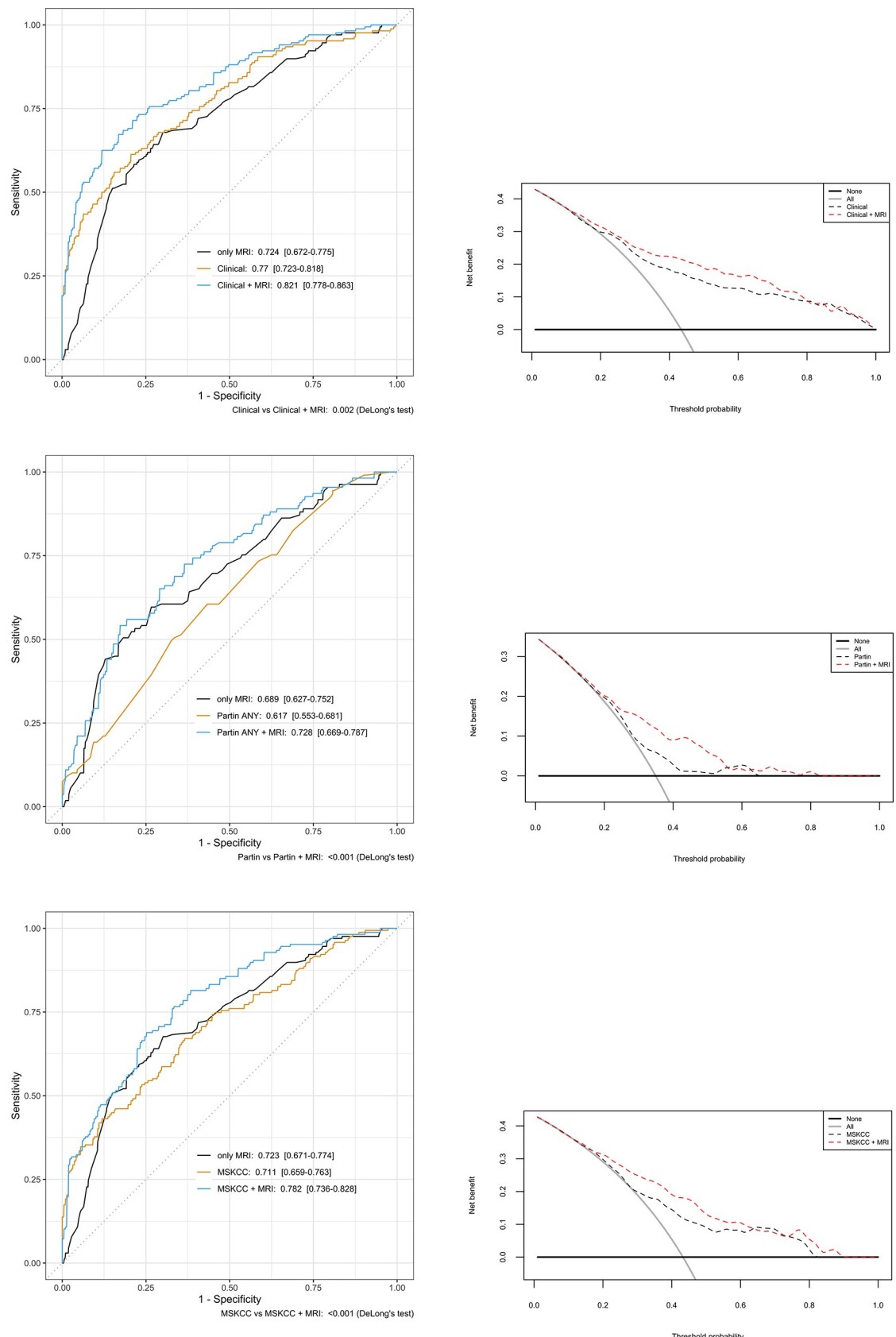

**Fig 1.** Additive predictive value of any adverse RP findings by mpMRI in ROC-AUC and Decision curve analysis (DCA) for a-b) clinical characteristics; c-d) Partin table estimates; e-f) MSKCC nomogram parameters.

In the Kaplan-Meier analysis, CAPRA risk score separated the patients into three distinct groups (Fig 2A–2C). In the CAPRA dataset, the time for BCR was significantly shorter when the mpMRI indicates adverse RP findings, than for the patients with favorable mpMRI findings. Additionally, for the high risk CAPRA group, mpMRI adverse findings separates the patients into two groups with significantly different survival times. In the intermediate CAPRA risk group, there were no differences in the survival times based on mpMRI findings (S4 Fig). For the low risk groups, there were only two events, thus no comparisons could be made.

When we looked at the MSKCC 3-year survival probability, a cut-off value of 80% significantly separated the patients into two groups with different survival probabilities (Fig 3A). In the high risk group, the mpMRI for any adverse RP finding significantly added value, separating the group into two cohorts with significantly different survival curves (Fig 3B). In the low risk MSKCC group there were no differences in the survival times based on mpMRI findings (S5 Fig).

As a positive surgical margin is a strong predictor of BCR all survival analyses were also done by excluding patients with a positive surgical margin [24]. All described results remained significant except for patient age (> 65 years), where the p-value increased to 0.633.

The cox proportional hazard models showed that mpMRI features associating with adverse RP findings correlate with BCR in an univariate model (S2 Table). However, when comparing models with and without the mpMRI indicating for adverse RP findings, no significant differences between the models were observed in the multivariable Wald test.

## Discussion

Here, we have addressed the value of combining mpMRI with clinical parameters, MSKCC nomogram and Partin nomogram to predict EPE, SVI or NS as adverse findings in RP in a contemporary cohort of PC patients. We can conclude that mpMRI improves the prediction of adverse pathology at RP when added to traditional preoperative prediction tools over a wide range of decision thresholds. Furthermore, mpMRI significantly added to the prediction of BCR when combined with the commonly used predictive tools CAPRA and MSKCC nomograms.

Our results on mpMRI-based prediction of adverse pathology at RALP are mostly in concordance with the literature. In a meta-analysis by de Rooij et al., a total of 75 studies (9796 patients) were analyzed and pooled for EPE (45 studies, 5681 patients), SVI (34 studies, 5677 patients), and overall stage T3 detection (38 studies, 4001 patients). The analysis for MRI found sensitivities and specificities for EPE, SVI and overall stage T3 detection of 0.57 and 0.91; 0.58 and 0.96; and 0.61 and 0.88, respectively. Functional imaging in addition to T2-weighted imaging and use of higher field strengths (3T) improved sensitivity for EPE and SVI. However, the heterogeneity of the studies included in the meta-analysis and the variability of the reported techniques (including pre PI-RADS era studies) hamper the interpretation of the results [25].

Recently, Morlacco et al. similarly evaluated the incremental value of MRI in a pre PI-RADS era (2003 to 2013) prostatectomy cohort [7]. The authors evaluated 914 patients with preoperative MRI of which 501 patients with endorectal coil MRI were eventually included in their analysis. They conclude that MRI provides added staging value and they also speculate that the inherent limitations in their study (pre PI-RADS era and high-risk patient cohort) may

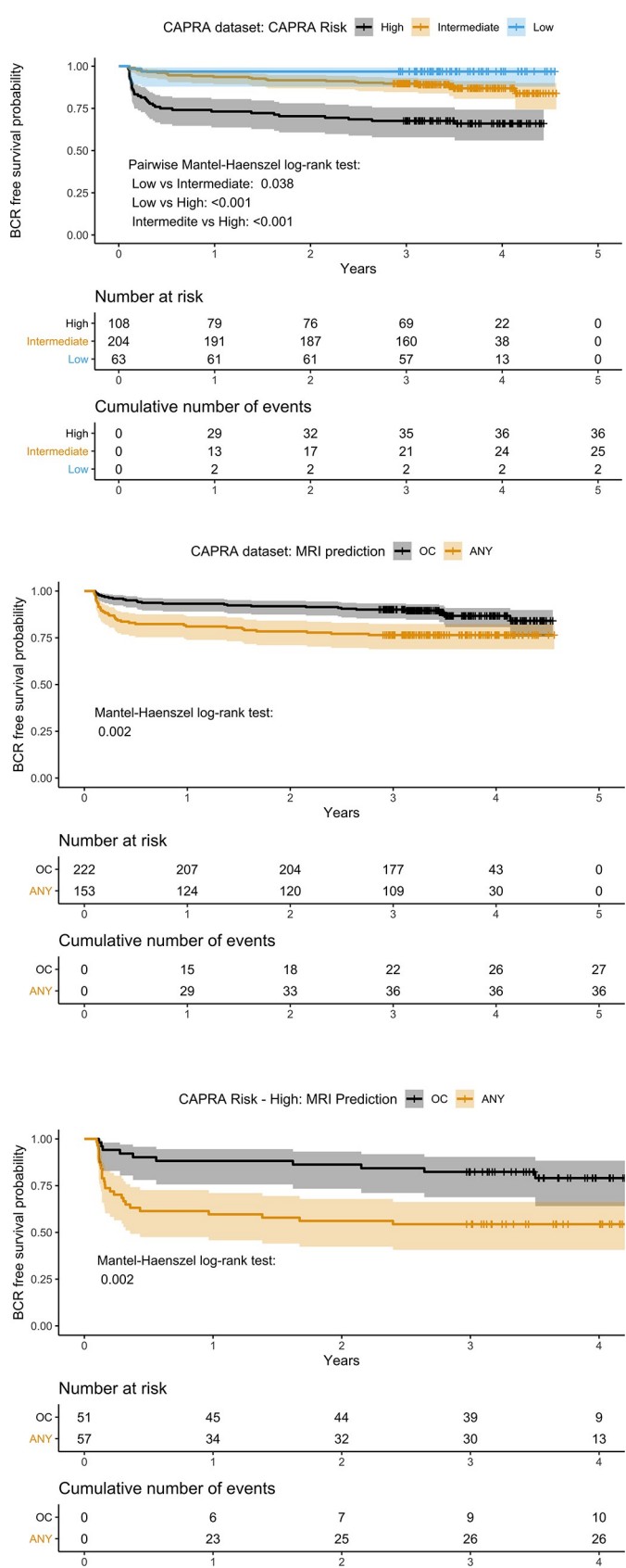

**Fig 2.** BCR free survival probability–a) Kaplan-Meier survival analysis for CAPRA risk groups; b) Survival plots for mpMRI any adverse finding (ANY) versus organ confined (OC) in the CAPRA group; c) Survival plot for mpMRI any adverse finding (ANY) versus organ confined (OC) in the CAPRA high risk group.

actually underestimate the added value of MRI. Our results suggest added value for MRI as a staging tool in the PI-RADS era. The study cohorts are relatively similar in respect to the risk factors (e.g. ≥pT3 49% vs 44%) but they substantially differ in MRI technique used (1.5T MRI and endorectal coil versus 3T pelvic coil and non-structured vs structured reporting). Furthermore, it is not possible to analyze the extent of selection bias in the Morlacco study as they did not report the characteristics of the cohort that did not undergo MRI [7]. Thus, one may speculate that clinical decision making was skewed towards guiding patients to MRI that were more likely to benefit from it. In addition to the AUC analysis, our decision curve analysis data suggested that the added net benefit for MRI extends over a wide range of threshold probabilities for EPE and SVI. Interestingly, another cohort with endorectal coil and no PI-RADS reported added value of preoperative MRI for staging of SVI [12]. The authors of that study reported significantly higher AUC values for Partin residues to predict SVI with and without MRI of 0.84 and 0.93, respectively, when compared to our data (0.68 and 0.73, respectively), suggesting different background risk in the two cohorts.

Gupta et al [26] addressed the accuracy of mpMRI and Partin Tables to predict EPE, but did not analyze the added value of mpMRI. In their analysis of only 60 patients, the ability of Partin Tables to predict EPE was comparable to ours (AUC of 0.62 vs 0.56). The mpMRI in their analysis, on the other hand, was a clearly better predictor of EPE than in our analysis (AUC of 0.82 vs 0.62). Significantly better predictive improvements have also been reported for MRI, when added to the Partin clinical model. In a subsequent study in 2016 Gupta et al found no added benefit of substituting multiparametric MRI stage for clinical stage (AUC of 0.63) to define organ-confined (OC) disease when using Partin tables (AUC of 0.70), while mpMRI was valuable as a stand-alone staging test (AUC 0.88) [27]. AUCs reported by Feng et al. for Partin and MSKCC nomograms predicting EPE were 0.85 and 0.86 and increased to 0.92 and 0.94, respectively, when mpMRI was added to each nomogram [28]. The Feng et al. and Gupta et al. studies relied on only one radiologist in their respective protocols to report the MRI results, which diminishes the effect of interreader variability in the results of both studies [26–28].

Our data suggests that when added to well-known prediction tools, prostate mpMRI aids in predicting BCR after RP, as it supports a more specific classification within the high risk CAPRA and MSKCC groups. Even as a standalone investigation, mpMRI was able to separate two groups with distinct BCR survival. Our results are in concordance with a small pre-PI-RADS era study where the ability of mpMRI to predict BCR after RP was seen, while the added value to CAPRA or MSKCC was not assessed [29]. As mpMRI is in part a surrogate method to predict RP findings, standardization of mpMRI reporting is expected to further improve its predictive value, as it is well known that adverse RP findings can predict BCR.

PI-RADS version 1 was the recommended reporting platform at the time of our study. PI-RADS version 2 or 2.1 may improve the inter-observer variability and diagnostic reliability [30]. Instead of a more recent PIRADS 2/2.1 cohort, we chose to analyze a PIRADS 1 cohort with longer follow-up time, which allowed us to use BCR as a study end point. The experience of the four radiologists in our study may be advantageous in diminishing inter-observer variability, especially for tumors in the peripheral zone as these are less dependent on the PI-RADS [31].

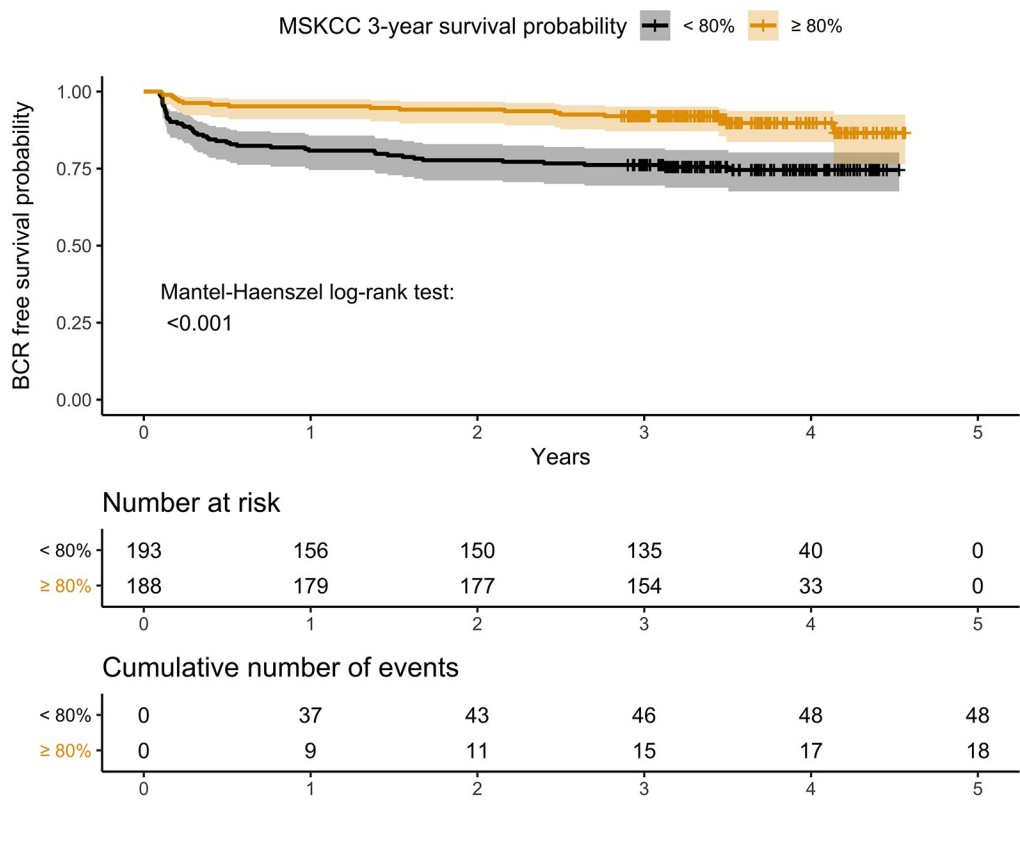

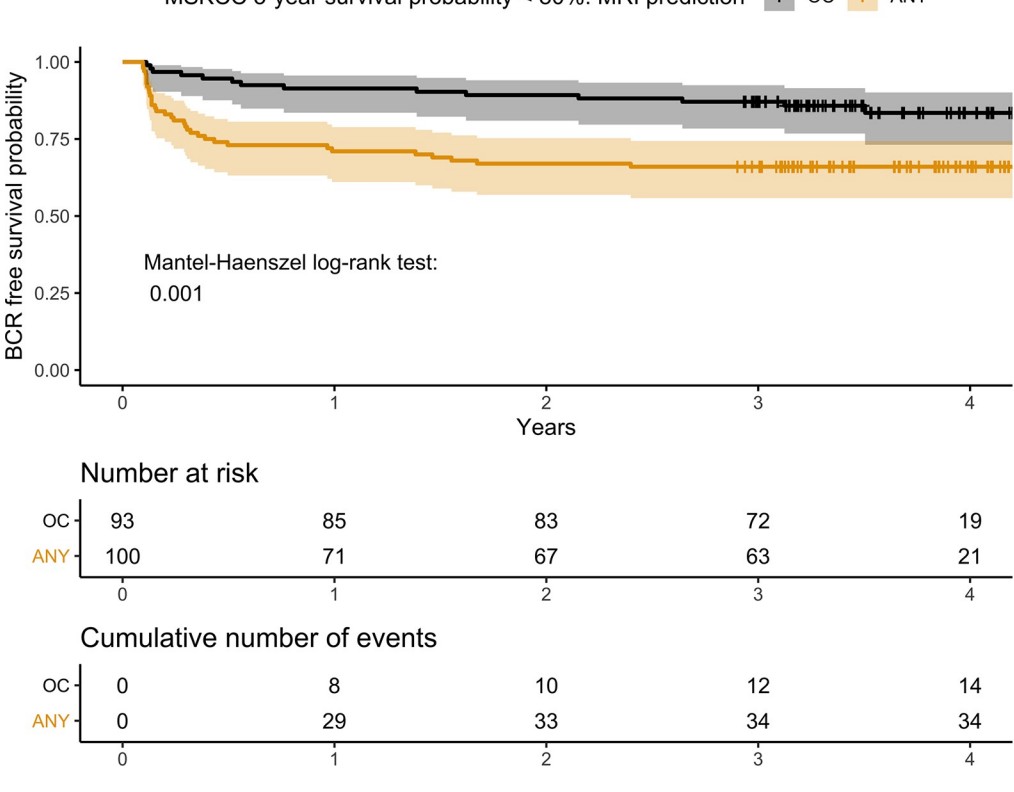

**Fig 3.** BCR free survival probability–a) Kaplan-Meier survival analysis for MSKCC BCR free survival at 80% after 3 years; b) Survival plot for mpMRI any adverse finding (ANY) versus organ confined (OC) in the MSKCC BCR high risk group (<80%).

Taken together, the evidence in the current literature tend to support the use of preoperative MRI as a staging tool, especially in regard to EPE and SVI. Despite the uncertainty of exactly predicting the presence of EPE or SVI, all the information combined, i.e. the size and location of the tumor among many other MRI and clinical variables may influence preoperative decision making and planning of the surgical procedure. This was recently shown in a retrospective cohort in which patients that had preoperative MRI were more likely to undergo non-nervesparing RP and had lower likelihood of positive surgical margins [24].

While the literature and the results by us may support the role of MRI as a staging tool for EPE and SVI preoperatively, the case is less clear for nodal status where a pooled sensitivity of only 0.39 has been reported [32]. A significant proportion of lymph node metastases are microscopic and enlarged lymph nodes may be reactive thereby hampering the size criterion in nodal status evaluation with any imaging modality, including MRI. Contrary to our results, recently a nomogram incorporating MRI has been developed to predict nodal status before RP [33]. This nomogram does not rely on the evaluation of the nodal status *per se* by MRI (contrary to our analysis). Instead, several MRI related variables are added to the existing model. This further supports the use of MRI preoperatively despite the actual evaluation of nodal status seems to be suboptimal with the current MRI techniques.

The limitations of our study include retrospective analysis of single center data. Selection bias cannot be ruled out as not all patients had preoperative MRI during the study period. However, no obvious differences where noted when we compared the demographics of these groups. Furthermore, in order to enhance diagnostic accuracy, to improve translational PC research and to better integrate MRI with pathology and to support clinical decision making, pathologists are encouraged to adopt regional reporting system similar to the PI-RADS [34].

Preoperative prostate MRI adds to prediction of adverse findings at RP and BCR after RP. As mpMRI is more commonly used for risk stratification at prebiopsy setting, mpMRI parameters are becoming more readily available and should be considered as additional source of information to the standard predictive nomograms.

## Supporting information

**S1 Table.** a) Comparison of mpMRI, clinical and pathological staging. Distribution of lymph node finding; 1 b) Distribution of PI-RADS and prostatectomy grading. PI-RADS score in multiparametric MRI and Gleason Grade Group in surgical specimen for ROI1.
(DOCX)

**S2 Table. Cox proportional hazard model summaries: prediction of biochemical recurrence.**
(DOCX)

**S1 Fig. Flowchart for the analyses.** LND: Lymph node dissection. CAPRA = Cancer of the Prostate Risk Assessment; MRI = magnetic resonance imaging.
(TIFF)

**S2 Fig.** BCR free survival probability–Kaplan-Meier survival analysis for clinical parameters: a) clinical T-stage equal or higher than 3 (follow-up data for clinical T-stage missing for one patient); b) Gleason Grade Group equal or higher than 3; c) preoperative PSA equal or higher than 10 µg/l; d) percentage of positive biopsies equal or higher than 50%; e) MRI prediction

for any adverse finding; f) age at operation equal or higher than 65 years.
(TIF)

**S3 Fig.** BCR free survival probability–Kaplan-Meier survival analysis for clinical parameters: a) mpMRI prostate volume ($>$ 35 cc); b) PI-RADS score higher than two.
(TIF)

**S4 Fig. BCR free survival probability–Kaplan-Meier survival analysis for intermediate CAPRA risk group.**
(TIFF)

**S5 Fig. BCR free survival probability–Kaplan-Meier survival analysis for low risk MSKCC group.**
(TIFF)

## Author Contributions

**Conceptualization:** Kevin Sandeman.

**Data curation:** Kevin Sandeman, Joona Pohjonen, Andrew Erickson, Suvi Marjasuo, Anu Kenttämies.

**Formal analysis:** Kevin Sandeman, Joona Pohjonen, Andrew Erickson.

**Funding acquisition:** Tuomas Mirtti, Antti Rannikko.

**Project administration:** Tuomas Mirtti.

**Resources:** Tuomas P. Kilpeläinen, Petrus Järvinen, Henrikki Santti, Anssi Petas, Mika Matikainen.

**Supervision:** Tuomas Mirtti, Antti Rannikko.

**Validation:** Antti Rannikko.

**Visualization:** Kevin Sandeman.

**Writing – original draft:** Kevin Sandeman, Juho T. Eineluoto.

**Writing – review & editing:** Kevin Sandeman, Juho T. Eineluoto, Tuomas Mirtti, Antti Rannikko.

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
