## [Decision Letter · Decision Letter 0]

19 May 2020

PONE-D-20-03900

Prostate MRI added to CAPRA, MSKCC and Partin cancer nomograms significantly enhances the prediction of adverse findings and biochemical recurrence after radical prostatectomy

PLOS ONE

Dear Dr. Sandeman,

Thank you for submitting your manuscript to PLOS ONE. After careful consideration, we feel that it has merit but does not fully meet PLOS ONE’s publication criteria as it currently stands. Therefore, we invite you to submit a revised version of the manuscript that addresses the points raised during the review process.

We would appreciate receiving your revised manuscript by Jul 03 2020 11:59PM. To enhance the reproducibility of your results, we recommend that if applicable you deposit your laboratory protocols in protocols.io, where a protocol can be assigned its own identifier (DOI) such that it can be cited independently in the future. For instructions see: http://journals.plos.org/plosone/s/submission-guidelines#loc-laboratory-protocols

We look forward to receiving your revised manuscript.

Kind regards,

Isaac Yi Kim, MD, PhD

Academic Editor

PLOS ONE

2. In the ethics statement in the manuscript and in the online submission form, please provide additional information about the patient records used in your retrospective study, including: a) whether all data were fully anonymized before you accessed them; b) the date range (month and year) during which patients' medical records were accessed; and c) the source of the medical records analyzed in this work (e.g. hospital, institution or medical center name). If patients provided informed written consent to have data from their medical records used in research, please include this information.

3. Please confirm in your methods section and ethics statement that the diary number stated in the ethics statement refers to the approval number issued by the ethics committee.

4. Please provide a sample size and power calculation in the Methods, or discuss the reasons for not performing one before study initiation.

"This study has been funded by Cancer Society Finland (TM, AR), Academy of Finland (TM), and HUS Helsinki University Hospital (TM, AR)."

"JE Orion Research Foundation sr

7. We note that you have included the phrase “data not shown” in your manuscript. Unfortunately, this does not meet our data sharing requirements. PLOS does not permit references to inaccessible data. We require that authors provide all relevant data within the paper, Supporting Information files, or in an acceptable, public repository. Please add a citation to support this phrase or upload the data that corresponds with these findings to a stable repository (such as Figshare or Dryad) and provide and URLs, DOIs, or accession numbers that may be used to access these data. Or, if the data are not a core part of the research being presented in your study, we ask that you remove the phrase that refers to these data.

Reviewers' comments:

Reviewer's Responses to Questions

**Comments to the Author**

1. Is the manuscript technically sound, and do the data support the conclusions?

Reviewer #1: Yes

2. Has the statistical analysis been performed appropriately and rigorously? 

Reviewer #1: Yes

3. Have the authors made all data underlying the findings in their manuscript fully available?

Reviewer #1: Yes

4. Is the manuscript presented in an intelligible fashion and written in standard English?

Reviewer #1: Yes

5. Review Comments to the Author

Reviewer #1: This is a nice retrospective study that among other studies is trying to establish what role mpMRI has in pre-treatment evaluation in men with prostate cancer. While there are many studies on this subject, many of which are referred to in the manuscript, I do think that having the experience of multiple centeres during this growing phase of mpMRI in prostate cancer evaluation is necessary to inform future necessary randomized controlled trials.

Specific questions I have:

- I understand that perhaps this was not a question the authors were addressing in this manuscript, and perhaps could be reserved for another study, but why did they not include positive surgical margin in their analysis? I would say many surgeons try to rely on preoperative MRI to guide their resection, and PSM would be an "adverse pathologic feature." Again, if the authors intent was simply to see the effect on established preoperative nomograms then I understand the exlusion, but I do feel it would be a simple addition to the paper and very helpful

- The authors do not explicitly state what their defintion of "adverse pathologic features" was. EPE, SVI, and node positive disease are inferred, but this is not explicity stated and I think needs to be clarified.

-While all MRIs were read my 4 qualified radiologists, can the authors clarify if the pathology readings were centralized by a single or small group of pathologists?

- I do not know if it is the formatting of the pdf file, but the tables are listed in the middle of the manuscript while the figures (without description) are listed at the end. This did make the manuscript difficult to follow.

6. PLOS authors have the option to publish the peer review history of their article (what does this mean?). If published, this will include your full peer review and any attached files.

Reviewer #1: Yes: Keith J. Kowalczyk

---

## [Author Response · Author response to Decision Letter 0]

10 Jun 2020

Rebuttal letter

PONE-D-20-03900

Response: We have reviewed and corrected the manuscript to match the PLOS ONE style requirements.

2. In the ethics statement in the manuscript and in the online submission form, please provide additional information about the patient records used in your retrospective study, including: a) whether all data were fully anonymized before you accessed them; b) the date range (month and year) during which patients' medical records were accessed; and c) the source of the medical records analyzed in this work (e.g. hospital, institution or medical center name). If patients provided informed written consent to have data from their medical records used in research, please include this information.

Response: The suggested changes have been made to the manuscript.

Revised manuscript (Materials and methods: Patient population. Sentences added): The data was accessed and collected between October 2016 and June 2019 from the electronic health records of the HUS Helsinki University Hospital. The original data was accessed based on patients’ social security numbers but in the study database all data was handled pseudonymized and according to the study approval by the HUS Helsinki University Hospital. The study was a retrospective registry study and thus no informed consent was required, based on the national and European Union legislation. The study was conducted according to the Declaration of Helsinki and approved by the Helsinki University Hospital ethics committee (diary number 386/13/03/02/2014).

3. Please confirm in your methods section and ethics statement that the diary number stated in the ethics statement refers to the approval number issued by the ethics committee.

Response: The diary number has been checked and assured to refer to the approval number.

4. Please provide a sample size and power calculation in the Methods, or discuss the reasons for not performing one before study initiation.

Response: Study sample size is addressed in the chapter Materials and methods: Patient population. The study included retrospective data on 387 consecutive radical prostatectomy patients that were diagnosed according to the contemporary multiparametric MRI and had a reasonably long follow-up time. Therefore, no formal power calculations were performed but, instead, all patients during the study period were included.

Revised manuscript (Materials and methods: Patient population. Sentences added): At the time of the data curation, we included all the patients having undergone MRI according to contemporary guidelines and having follow-up information in order to conduct survival analysis. Therefore, no formal power calculations were performed but, instead, all patients during the study period were included.

Response: The ORCID ID of the corresponding author has been updated.

6. Thank you for stating the following in the Acknowledgments Section of your manuscript: "This study has been funded by Cancer Society Finland (TM, AR), Academy of Finland (TM), and HUS Helsinki University Hospital (TM, AR)." We note that you have provided funding information that is not currently declared in your Funding Statement. However, funding information should not appear in the Acknowledgments section or other areas of your manuscript. We will only publish funding information present in the Funding Statement section of the online submission form. Please remove any funding-related text from the manuscript and let us know how you would like to update your Funding Statement. Currently, your Funding Statement reads as follows:

"JE Orion Research Foundation sr The funders had no role in study design, data collection and analysis, decision to publish, or preparation of the manuscript."

Response: We have removed the funding information from the Acknowledgments section and updated the Funding Statement in the online submission form.

7. We note that you have included the phrase “data not shown” in your manuscript. Unfortunately, this does not meet our data sharing requirements. PLOS does not permit references to inaccessible data. We require that authors provide all relevant data within the paper, Supporting Information files, or in an acceptable, public repository. Please add a citation to support this phrase or upload the data that corresponds with these findings to a stable repository (such as Figshare or Dryad) and provide and URLs, DOIs, or accession numbers that may be used to access these data. Or, if the data are not a core part of the research being presented in your study, we ask that you remove the phrase that refers to these data.

Response: The suggested changes have been made to the manuscript and we have added the data previously not shown as supplementary files.

Revised manuscript (Results: BCR free survival (CAPRA, MSKCC, clinical variables, mpMRI). Sentence corrected):

Neither mpMRI prostate volume (> 35 cc) nor the PIRADS score higher than two separated the groups significantly (Fig S3 a-b).

Previous manuscript:

Neither mpMRI prostate volume (> 35 cc) nor the PIRADS score higher than two separated the groups significantly (data not shown).

Revised manuscript (Results: BCR free survival (CAPRA, MSKCC, clinical variables, mpMRI):

In the intermediate CAPRA risk group, there were no differences in the survival times based on mpMRI findings (Fig S 4). For the low risk groups, there were only two events, thus no comparisons could be made.

Previous manuscript:

In the low and intermediate CAPRA risk groups, there were no differences in the survival times based on mpMRI findings (data not shown).

Revised manuscript (Results: BCR free survival (CAPRA, MSKCC, clinical variables, mpMRI). Sentence corrected):

In the low risk MSKCC group there were no differences in the survival times based on mpMRI findings (Fig S 5).

Previous manuscript:

In the low risk MSKCC group there were no differences in the survival times based on mpMRI findings (data not shown).

Response: Suggested corrections were made according to the Supporting Information guidelines.

Revised manuscript (Supporting information added):

Supporting information

S1 Table. Supplementary table 1 a) Comparison of mpMRI, clinical and pathological staging. Distribution of lymph node finding; 1 b) Distribution of PI-RADS and prostatectomy grading. PI-RADS score in multiparametric MRI and Gleason Grade Group for ROI1.

S2 Table. Supplementary table 2. Cox proportional hazard model summaries: prediction of biochemical recurrence.

S1 Fig. Flowchart for the analyses. LND: Lymph node dissection. CAPRA = Cancer of the Prostate Risk Assessment; MRI = magnetic resonance imaging.

S2 Fig. BCR free survival probability – Kaplan-Meier survival analysis for clinical parameters: a) clinical T-stage equal or higher than 3 (follow-up data for clinical T-stage missing for one patient); b) Gleason Grade Group equal or higher than 3; c) preoperative PSA equal or higher than 10 µg/l; d) percentage of positive biopsies equal or higher than 50%; e) MRI prediction for any adverse finding; f) age at operation equal or higher than 65 years . 

S3 Fig. BCR free survival probability – Kaplan-Meier survival analysis for clinical parameters: a) mpMRI prostate volume (> 35 cc); b) PIRADS score higher than two.

S4 Fig. BCR free survival probability – Kaplan-Meier survival analysis for low risk MSKCC group.

S5 Fig. BCR free survival probability – Kaplan-Meier survival analysis for low risk MSKCC group.

2. Reviewers’ Comments to the Author

1. I understand that perhaps this was not a question the authors were addressing in this manuscript, and perhaps could be reserved for another study, but why did they not include positive surgical margin in their analysis? I would say many surgeons try to rely on preoperative MRI to guide their resection, and PSM would be an "adverse pathologic feature." Again, if the authors intent was simply to see the effect on established preoperative nomograms then I understand the exlusion, but I do feel it would be a simple addition to the paper and very helpful

Response: We thank the reviewer for this valuable comment. As stated, the PSM is not a part of the preoperative nomograms in this study. PSM is very much surgeon and hospital dependent and other adverse pathology findings in RP (i.e. EPE, SVI and LN) have been also shown to have impact on survival. However, the role and impact of preoperative MRI on surgical strategy (nerve sparing vs no nerve sparing), surgical margin status, and postoperative function are the scope of another separate ongoing analysis.

No corrections in the manuscript were made.

2. The authors do not explicitly state what their defintion of "adverse pathologic features" was. EPE, SVI, and node positive disease are inferred, but this is not explicity stated and I think needs to be clarified.

Response: We have now clarified the definition of adverse pathologic features better in the manuscript.

Revised manuscript (Materials and methods. Prostatectomy specimen and pathologic analysis. Sentence added): Adverse pathologic findings were defined as EPE, SVI or positive NS.

3. While all MRIs were read my 4 qualified radiologists, can the authors clarify if the pathology readings were centralized by a single or small group of pathologists?

Response: We had altogether 12 pathologists performing the pathology readings, and five of them performed over 80% of the readings. We have now better clarified this in the manuscript.

Revised manuscript (Materials and methods. Prostatectomy specimen and pathologic analysis. Sentence added): The original pathological diagnosis was made individually by 12 expert pathologists, out of which 4 reported 80% of the cases. 

4. I do not know if it is the formatting of the pdf file, but the tables are listed in the middle of the manuscript while the figures (without description) are listed at the end. This did make the manuscript difficult to follow.

Response: The manuscript has now been carefully reviewed and corrected to match the PLOS ONE style requirements.

---

## [Editor Report · Decision Letter 1]

23 Jun 2020

Prostate MRI added to CAPRA, MSKCC and Partin cancer nomograms significantly enhances the prediction of adverse findings and biochemical recurrence after radical prostatectomy

PONE-D-20-03900R1

Dear Dr. Sandeman,

We’re pleased to inform you that your manuscript has been judged scientifically suitable for publication and will be formally accepted for publication once it meets all outstanding technical requirements.

Kind regards,

Isaac Yi Kim, MD, PhD

Academic Editor

PLOS ONE
---

## [Editor Report · Acceptance letter]

26 Jun 2020

PONE-D-20-03900R1 

Prostate MRI added to CAPRA, MSKCC and Partin cancer nomograms significantly enhances the prediction of adverse findings and biochemical recurrence after radical prostatectomy 

Dear Dr. Sandeman:

I'm pleased to inform you that your manuscript has been deemed suitable for publication in PLOS ONE. Congratulations! Your manuscript is now with our production department. 

Kind regards, 

on behalf of

Dr. Isaac Yi Kim 

Academic Editor

PLOS ONE